# Meniscal Tear Outcome (METRO) review: a protocol for a systematic review summarising the clinical course and patient experiences of meniscal tears in the current literature

Imran Ahmed [1], Chetan Khatri,[2] Nicholas Parsons,[3] Charles E Hutchinson,[3] Sophie Staniszewska,[3] Andrew James Price [1],[4] Andrew Metcalfe [1]

¹Clinical Trials Unit, University of Warwick, Coventry, UK
²Trauma and Orthopaedics, --University Hospital Coventry and Warwickshire, Coventry, UK
³Warwick Medical School, University of Warwick, Coventry, UK
⁴Nuffield Department of Orthopaedics, Rheumatology and Musculoskeletal Sciences, University of Oxford, Oxford, UK

**Correspondence to**
Imran Ahmed;
imran.ahmed4@nhs.net

## ABSTRACT

**Introduction** Meniscal tears are a common knee injury with an incidence of 60 per 100 000. Management of meniscal tears can include either non-operative measures or operative procedures such as arthroscopic partial meniscectomy (APM). Despite substantial research evaluating the effectiveness of APM in the recent past, little is known about the clinical course or the experiences of patients with a meniscal tear.

**Aim** To summarise the short to long-term patterns of variability in outcome in patients with a meniscal tear. To summarise the evidence on patient experiences of meniscal tears. In particular, we will focus on patient experiences of treatment options, treatment pathways and their views of the outcomes used in meniscal tear research.

**Methods and analysis** Two search strategies will be developed to identify citations from EMBASE, MEDLINE, AMED, CENTRAL, Web of Science and Sociofile. The date of our planned search is 14 August 2020. For the quantitative review we will identify studies reporting patient-reported outcome measures in patients after a meniscal tear. The standardised mean change will be used to assess the variation in size of response and summarise the overall response to each treatment option. All studies will undergo quality assessment using either the Cochrane risk of bias or the Newcastle-Ottawa tool.

A qualitative systematic review will be used to identify studies reporting views and experiences of patients with a meniscal tear. All studies will be assessed using the Critical Appraisal Skills Programme tool and if sufficient data are present a meta-synthesis will be performed to identify first, second and third-order constructs.

**Ethics and dissemination** Given the nature of this study, no formal ethical approval will be sought. Results from the review will be disseminated at national conferences and will be submitted to a peer-reviewed journal for publication. Lay summaries will be freely available via the study Twitter page.

**PROSPERO registration number** CRD42019122179.

### Strengths and limitations of this study

- ► Comprehensive nature of review including patients undergoing all management options.
- ► Analysis plan in keeping with previously published systematic reviews.
- ► Risk of publication bias as only English language studies are to be included.
- ► Patient and public involvement from study conception through to planned dissemination of results.

## INTRODUCTION

Meniscal tears are a common knee injury with an annual incidence of 60–70 per 100 000 and they account for 70 000 hospital admissions per year in the UK.[1 2] The menisci fibrocartilaginous structures are situated within the knee.[3] The principal role of the menisci is to distribute compressive load within the knee joint by increasing the contact area and reducing the contact pressure between the tibia and the femur.[3–5] This may play a crucial role in preventing osteoarthritis (OA). Risk factors for meniscal tears include age, gender, body mass index (BMI), presence of degeneration within the knee and level of activity, for example, participation in sport.[3] Meniscal tears can be asymptomatic or present with symptoms such as knee pain, swelling, stiffness or locking of the knee[3 6 7]; however, it is unknown if these symptoms are caused by the meniscal tear specifically.

The management options for treatment of meniscal tears include watchful waiting (observation), medical therapy (intra-articular injections, analgesia), exercise therapy and surgery.[8–11] Recent evidence has questioned the effectiveness of surgery when compared with exercise therapy or other non-operative interventions.[12–14] Despite this evidence, the number of arthroscopies has almost doubled over the last 15 years.[15 16] Many surgeons and researchers believe surgery may be beneficial

BMJ

in a subset of patients and over recent years there has been a substantial growth in the number of studies and systematic reviews in the management of meniscal tears.[17–19] These studies are yet to identify a clear superior treatment option in this patient population, or to identify a true subset that may benefit from surgery.

Research into other musculoskeletal conditions has demonstrated that outcomes steadily improve over time regardless of treatment.[20 21] This could be due to the natural history of the condition or simply regression to the mean. The natural history of a condition is a crucial factor to consider during interpretation of outcomes from interventions of patients with meniscal tears. Prospective studies with well-defined entry criteria and follow-up points provide an excellent source of information on the natural history of a condition.[20] Researchers and clinicians need to better understand the natural history of meniscal tears to plan treatment decisions and future research.

In addition to understanding the clinical course of patients being managed with a meniscal tear, we need to understand patient views and experiences of having and being treated for meniscal tears, as a form of patient-based evidence which we use alongside clinical and economic forms of evidence.[22] Recent treatment guidelines were developed, based on the consensus of 25 international experts, by the British Association for Surgery of the Knee.[11] However, no patients were involved in the development of these guidelines. In order to treat meniscal tears more effectively and plan informative research we need to understand patient views and expectations in the treatment of meniscal tears.

## Objectives
### Quantitative review
The objective of the quantitative review is to summarise the short, medium and long-term temporal patterns of change in patient-reported outcome measures (PROM) in patients with a meniscal tear regardless of treatment.

### Qualitative review
The objective of the qualitative review is to identify and summarise the evidence on patient experiences of meniscal tears. In particular we will focus on patient experiences of treatment options (surgical and non-surgical), treatment pathways and their views of the outcomes used in meniscal tear research.

## METHODS AND ANALYSIS
This study protocol was developed in accordance with the Preferred Reporting Items for Systematic Review and Meta-Analysis Protocols checklist. In addition to this manuscript the protocol has been registered on the PROSPERO database and can be found at https://www.crd.york.ac.uk/PROSPERO/display_record.php?RecordID=122179

## Eligibility criteria
### Quantitative review
*Inclusion criteria*
► Study design
  – Full-text randomised controlled trials (RCT) in patients treated for a meniscal tear (eg, comparing surgery or a non-operative intervention (exercise therapy, pharmacological therapy or observation) vs a comparator group).
  – Prospective cohort studies adjusted for case mix reporting outcomes in patients being treated for a meniscal tear.
► Studies reporting at least one established knee-related PROM score for at least 3 months.
► English language studies only.

*Exclusion criteria*
► Studies not reporting any of the clinical outcomes selected for this review.
► Studies reporting outcomes in patients with additional major knee ligament injury or fractures around the knee.
► Abstract or conference publications.

### Qualitative review
*Inclusion criteria*
► Qualitative studies reporting the views and experiences of patients with a meniscal tear, implementing all types of qualitative methodology and method.
► English language studies only.

*Exclusion criteria*
► Abstract or conference publications.

## Search strategy and quality assessment
Following consultation with a librarian a search strategy was devised (see online supplementary material). The authors aim to search the following databases: MEDLINE; Excerpta Medica Database (EMBASE); Allied and Complementary Medicine (AMED); and Cochrane Central Register of Controlled Trials (CENTRAL) using OVID Sp. We will also search Web of Science and Sociofile for qualitative studies. Search strategies can be seen in the online supplementary appendices 1 and 2. We will also search reference lists of included studies to identify further citations. All citations will be imported into EndNote X9 reference management software[23] (Clarivate Analytics, Philadelphia, USA). After removal of duplicates, titles and abstracts will be screened according to the inclusion criteria. The full texts of studies potentially meeting the inclusion criteria will be screened by two authors (IA and CK). These authors will independently assess each study and any discrepancies will be addressed by discussion with a senior author (NP or AM).

RCTs will be qualitatively assessed using the Cochrane risk of bias tool.[24] Observational studies will be assessed using the Newcastle-Ottawa Scale.[25] Two authors (IA and CK) will independently assess the quality of each included study with discrepancies being resolved following discussion with a senior author (NP or AM).

Each qualitative study will be critically appraised using the Critical Appraisal Skills Programme for qualitative studies.[26] An in-depth quality assessment of each qualitative study will be performed accounting for validity of the results, appropriateness of the results and the applicability of the results. Each study will be graded as adequate, partially adequate or inadequate. Grading will be performed by two review authors (IA and CK) and any discrepancies will be discussed with a senior author (AM). If rated as inadequate the study will be excluded from the meta-synthesis.

Two authors (IA and CK) will independently assess the quality of evidence. We will use the Grading of Recommendations Assessment, Development and Evaluation (GRADE) considerations (study limitations, consistency of effect, imprecision, indirectness and publication bias) to assess the quality of the body of evidence.[27] Decisions to upgrade or downgrade body of evidence will be clearly stated.

## Outcomes
### Primary outcome
The primary outcome measure is knee pain or function measured by at least one of the following validated knee questionnaires: (1) Knee Injury and Osteoarthritis Outcome Score; (2) Oxford Knee Score; (3) Western Ontario Meniscal Evaluation Tool; (4) Lysholm score; (5) International Knee Documentation Committee score; and (6) Western Ontario and McMaster Universities Osteoarthritis Index score.

## Data extraction
### Extraction plan for quantitative studies
For each study, we will collect the number of patients (in each arm if RCT or non-randomised study) and the intervention type, defined as meniscectomy, meniscal repair, placebo surgery or non-operative. In addition, mean age, gender, BMI, method of diagnosis of tear, imaging findings (presence of OA), patient-reported symptoms, whether patient and public involvement (PPI) was reported and means and SDs of the PROM score at each time point reported will also be collected. We will also extract information on whether any qualitative data were included in the final study report. If a study does not report mean or SD, we will contact the study authors for further information. If the SD is not provided then where possible estimates will be calculated from other reported statistics (data summaries), for example, a test statistic or p value, using standard methods as described in the Cochrane Handbook.[24] If data are not represented in a simple numerical format (eg, in a table or in the study text), but are presented graphically only (eg, a figure or graph), two authors (IA and CK) will extract the data manually. If we fail this and we are unable to extract the data, we will contact the study authors and request the data.

### Data analysis
Outcome scores will be plotted against time from study intervention (eg, non-operative (exercise therapy or pharmacological therapy) or surgery) to describe change from baseline (at the point of recruitment) to all follow-up time points reported in the treatment arms included. This will provide a simple visual representation of the clinical course of the condition. If studies report an observation or a 'watch and wait' arm where no treatment is initiated we will plot this separately. This will provide an insight into the natural history of the condition.

To determine variation in size of response, the change of outcome will be assessed by calculating the bias-corrected standardised mean change (SMC) at 3, 6, 12, 24, 36, 48 and 60 months. The SMC is calculated by subtracting the follow-up mean score for the PROM of interest from the baseline mean score. This is then divided by the pooled SD and multiplied by the bias correction factor based on group size. Follow-up or baseline SD will be used if the pooled SD is not reported or provided when study authors are contacted for further information. The 95% CIs will be calculated using estimates of the variance of the SMC.[28]

Studies using a variety of outcome scales or PROMs can be pooled as the SMC standardises the measurement of change. As in previously reported meta-analyses we will calculate a combined pooled SMC for each time point using a random effects model.[20 21] We will do this for the meniscectomy arm, meniscal repair arms and non-operative arms of studies. A simple correlation analysis (using Pearson's correlation coefficient) on all SMCs between each pair of time points will be used to assess the relationship between intervals.

Should the data allow, we will perform a subgroup analysis based on presence or absence of OA and also based on mechanism of injury. Tears with a clearly defined history of trauma will be compared with those without a clear history of trauma, and tears associated with features of OA will be compared with tears without associated OA. This will allow exploration of change in outcome scores over time in the presence or absence of OA and trauma. Further subgroup analysis will be performed where we will pool PROMs for exercise therapy and pharmacological interventions separately. These will describe the clinical course of non-pharmacological and pharmacological interventions, where such data are available.

We will also conduct a separate analysis comparing outcomes for operative interventions versus non-operative interventions. A random effects model will be used to pool the standardised mean differences of comparable groups of studies in a meta-analysis. Examples of comparisons which will be made include: (1) arthroscopic partial meniscectomy (APM) versus non-operative interventions; (2) meniscal repair versus non-operative interventions; and (3) meniscal repair versus APM.

All analyses will be implemented in R[29] (https://www.r-project.org/) using the metafor package.[30]

### Extraction plan for qualitative studies
A prospective data extraction sheet will be designed to collect the following data: author, year of publication, journal, and aim of study, number of participants, data

collection methods, theoretical or methodological underpinnings and results.

### Analysis and meta-synthesis

If sufficient studies are identified, a meta-synthesis will be performed to translate existing evidence into new theory. Meta-synthesis is a process which involves synthesis of findings from qualitative studies, interpreting the results and generating new interpretations.[31] Each included study will be read and re-read to produce a table of first-order constructs (direct quotes from the participants in each individual study) and second-order constructs (author interpretation of the meaning of individual participant quotes). We will collect data (or quotes) on participant views on treatment options, treatment pathways and the current outcomes used in meniscal tear research. Thematic coding will be used to identify cluster of themes in order to develop third-order constructs (our interpretation of the cluster themes).

## DISCUSSION

Previous reviews have focused on comparing the effectiveness of surgical intervention (arthroscopic meniscectomy) versus a placebo or non-operative intervention.[12 17] There has been no review reporting the clinical course or natural history of the condition. This review will provide a comprehensive summary of the patient-reported outcomes of all patients being managed with a meniscal tear in the literature. Our analysis will provide an insight into the natural history of the condition and the patterns of change in PROMs following operative management. PPI in each of the included studies will also be explored.

There has been no qualitative synthesis in the literature of patient experiences of meniscal tears. This review will explore and summarise the available qualitative evidence on patient experiences of living with a meniscal tear. This will identify if there is an evidence gap which will require the need for further qualitative research in this patient population.

This review will inform future research through developing the understanding of the natural history of patients with a meniscal tear. It will also inform researchers and healthcare providers on patient views and experiences on living with a meniscal tear.

A potential weakness of this review is based on the studies included. Studies may not include an observation arm where there is no treatment initiated. As a result it will be difficult to describe the true natural history of patients with meniscal tears. Another weakness is that we are pooling the results of all patients with meniscal tears where there is the possibility of clinical heterogeneity in the sample. A recent cohort study reported factors such as age, gender and symptom duration did not influence treatment outcome.[18] As a result a decision was taken to group all tears together.

The strengths of the review include its comprehensive nature of its design. We will include patients being managed by both operative and non-operative measures.

Studies will be quality assessed using appropriate methodology and the body of evidence will be assessed using the GRADE criteria. A further strength of this piece of work is the incorporation of qualitative data. This will be the first review to summarise the evidence exploring participant views on treatment options and treatment pathways. This will provide further insight for clinicians in order to determine treatment decisions.

## PATIENT AND PUBLIC INVOLVEMENT

A PPI group of 18 patients with knee OA or meniscal tears was set up to discuss this study idea. Input was provided on the importance of patient-focused dissemination of the results and the use of lay summaries. A patient member (Dilshad Sachedina) reviewed the aspects of the manuscript, in particular the study abstract and dissemination plans. A reference group will be established with whom key findings of the study will be discussed and interpretations coproduced. A lay summary of the final results of the review will be produced in combination with the PPI reference team and we will make it freely available via a study Twitter page or email if required.

## ETHICS AND DISSEMINATION

Given the study design, no formal ethical approval will be sought to conduct this review. The findings will be disseminated at national and international conferences. The final report will be submitted for peer-reviewed publication. A lay summary will be produced and be freely available via the study Twitter page.

**Contributors** IA, NP, CEH, SS, AJP, AM: study conception, drafted and reviewed the final manuscript. CK: drafted and reviewed the final manuscript.

**Funding** This protocol represented research funded by the National Institute of Health Research (NIHR) Doctoral Research Fellowship grant awarded to IA (DRF-2018-11-ST2-030).

**Competing interests** None declared.

**Patient consent for publication** Not required.

**Provenance and peer review** Not commissioned; externally peer reviewed.

**ORCID iDs**
Imran Ahmed http://orcid.org/0000-0003-2774-9954
Andrew James Price http://orcid.org/0000-0002-4258-5866
Andrew Metcalfe http://orcid.org/0000-0002-4515-8202

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
