## [Reviewer comments · BMJ Open]

ARTICLE DETAILS

TITLE (PROVISIONAL)	The Meniscal tear outcome (METRO) review: A protocol for a systematic review summarising the clinical course and patient experiences of meniscal tears in the current literature.
AUTHORS	Ahmed, Imran; Khatri, Chetan; Parsons, Nicholas; Hutchinson, Charles; Staniszewska, Sophie; Price, Andrew; Metcalfe, Andrew

VERSION 1 – REVIEW

REVIEWER	Victor van de Graaf OLVG Amsterdam, the Netherlands
REVIEW RETURNED	22-Jan-2020

GENERAL COMMENTS	Thank you for the opportunity to review this manuscript. All in all a very interesting topic. There are several issues that need to be addressed: How will you account for different kind of meniscal tears? The way I read it, all specific meniscal treatments will be pooled despite the population. A meniscal tear in a young and active sports player will definitely behave differently than a degenerative one in a 70y old. What to do with the interpretability? And how will you deal with heterogeneity? Could the authors please explain: -why the WOMAC was not included as outcome measure?-why only outcomes up to 36 months are included? Why for instance exclude 60 months?-“Outcome scores will be plotted against time from study intervention (e.g. non-operative or surgery) to describe change from baseline (at the point of recruitment) to all follow up time points reported in the treatment arms included. This will provide a simple visual representation of the clinical course of the condition”- Could the authors elaborate on this? Why do you think that changed outcome scores after a gives treatment represent the clinical course of the condition? Also, how do you correct for placebo / nocebo effect? And since physical therapy is a treatment, how can you conclude on the natural history after PT? Please explain.- Then on page 10/17, line 14: “Follow up or baseline SD will be used if the pooled SD is not reported”- why don't you first contact the authors of the papers to retrieve this information? In my opinion, this will increase the reproducibility of the results.-The part about the qualitative evidence on patient experiences needs further explanation. It's unclear to me the scope of this part
---

	is, how this will be done, what specific data will be looked for, how this data will be analysed, etc. All in all, I believe more insight in the natural course of meniscal tears is really important. However, I'm not complete convinced how this review will provide an answer to this question. Hopefully the authors can explain in more detail. Finally, I recommended a statistical review, as I'm no expert in the field of standardised mean change (SMC).
--	--

REVIEWER	Kenneth Pihl University of Southern Denmark Denmark
REVIEW RETURNED	24-Feb-2020

GENERAL COMMENTS	Thank you for the opportunity to review this interesting protocol manuscript by Ahmed et al. entitled "The Meniscal tear outcome (METRO) review: A protocol for a systematic review summarising the natural history and patient experiences of meniscal tears in the current literature". The main aims of the protocolled systematic review are to summarize the natural history of symptoms in patients presenting with a meniscal tear and their experience of having a meniscal tear. The aims in focus are highly interesting given that meniscal tears are very common and a number of RCTs have failed to show surgery more effective than sham (placebo) surgery and exercise therapy, suggesting that improvements in symptoms may just be part of the natural time course and not necessarily a results of any treatment. The latter is a crucial factor to take into consideration when interpreting outcomes from interventions on people with meniscal tears. Overall, the protocol is well-written, the introduction builds a nice case for why the study is needed, and the methods are well-chosen. Besides one larger issue, I only have some minor comments. The coherence between the aims and the title and methods are a bit vague throughout the protocol. As understood from the title and the introduction, the aim is to look at the natural history of symptoms in patients with a meniscal tear. However, in the methods it is described that also patients receiving any kind of treatment will be included, which will not provide any knowledge of the natural history of symptoms, as these may be manipulated by the treatment. In line with this, the authors state that the patients having surgery will constitute the clinical course of the condition, whereas the non-operative treatment arm (e.g. physiotherapy, injections, etc.) will provide insight into the natural history of the condition. The latter implies that any non-operative treatment will not change the course of symptoms, which is a quite strong presumption. Only patients in a comparative arm (e.g. receiving placebo therapy) or receiving a wait-and-see approach can give insight into the natural course. It can even be discussed if another study looking into the clinical course of surgery, at least for arthroscopic partial meniscectomy, is needed as this has been reported in a number of recent systematic reviews.
---

	For the minor comments, see below: Introduction: In page 4, line 16-17 it is stated that meniscal tears can be symptomatic. Should be rephrased as even if symptoms, it is unknown if they are caused by the meniscal tear per se. In page 4 (and other places in the manuscript) line 21 treatment in form of “physiotherapy” is stated. Please elaborate what is mean by “physiotherapy” (e.g. exercise therapy, manual therapy, ultrasound, etc.) – Physiotherapy is a profession, not a specific treatment. Please provide a better reference than reference 15 for the number of surgeries. Reference 15 is a clinical guideline, not a study looking at the number of surgeries. Objectives: To the quantitative aim, “regardless of treatment/management” could be added to make it clearer. The aim for the qualitative study is not fully coherent with the aim in the abstract – the latter part from the abstract is missing. Methods: Page 7, line 3 – why written in past tense here? Discussion: A short discussion of the possible limitations of the study would be worthy.
--	---

VERSION 1 – AUTHOR RESPONSE

Thank you for the opportunity to review this interesting protocol manuscript by Ahmed et al. entitled “The Meniscal tear outcome (METRO) review: A protocol for a systematic review summarising the natural history and patient experiences of meniscal tears in the current literature”.

The main aims of the protocolled systematic review are to summarize the natural history of symptoms in patients presenting with a meniscal tear and their experience of having a meniscal tear. The aims in focus are highly interesting given that meniscal tears are very common and a number of RCTs have failed to show surgery more effective than sham (placebo) surgery and exercise therapy, suggesting that improvements in symptoms may just be part of the natural time course and not necessarily a results of any treatment. The latter is a crucial factor to take into consideration when interpreting outcomes from interventions on people with meniscal tears.

Overall, the protocol is well-written, the introduction builds a nice case for why the study is needed, and the methods are well-chosen. Besides one larger issue, I only have some minor comments.

The coherence between the aims and the title and methods are a bit vague throughout the protocol. As understood from the title and the introduction, the aim is to look at the natural history of symptoms in patients with a meniscal tear. However, in the methods it is described that also patients receiving any kind of treatment will be included, which will not provide any knowledge of the natural history of symptoms, as these may be manipulated by the treatment. In line with this, the authors state that the patients having surgery will constitute the clinical course of the condition, whereas the non-operative

treatment arm (e.g. physiotherapy, injections, etc.) will provide insight into the natural history of the condition. The latter implies that any non-operative treatment will not change the course of symptoms, which is a quite strong presumption. Only patients in a comparative arm (e.g. receiving placebo therapy) or receiving a wait-and-see approach can give insight into the natural course. It can even be discussed if another study looking into the clinical course of surgery, at least for arthroscopic partial meniscectomy, is needed as this has been reported in a number of recent systematic reviews.

For the minor comments, see below:

Introduction:

In page 4, line 16-17 it is stated that meniscal tears can be symptomatic. Should be rephrased as even if symptoms, it is unknown if they are caused by the meniscal tear per se.

In page 4 (and other places in the manuscript) line 21 treatment in form of “physiotherapy” is stated. Please elaborate what is mean by “physiotherapy” (e.g. exercise therapy, manual therapy, ultrasound, etc.) – Physiotherapy is a profession, not a specific treatment.

Please provide a better reference than reference 15 for the number of surgeries. Reference 15 is a clinical guideline, not a study looking at the number of surgeries.

Objectives:

To the quantitative aim, “regardless of treatment/management” could be added to make it clearer.

The aim for the qualitative study is not fully coherent with the aim in the abstract – the latter part from the abstract is missing.

Methods:

Page 7, line 3 – why written in past tense here?

Discussion:

A short discussion of the possible limitations of the study would be worthy.

VERSION 2 – REVIEW

REVIEWER	Victor van de Graaf OLVG Amsterdam, the Netherlands
REVIEW RETURNED	07-Apr-2020

GENERAL COMMENTS	Clear and satisfactory revision of the manuscript. I only have minor issue, that is for the interpretation of the results. As stated in my previous revision, how will the authors deal with the type of tear? Are you planning on subgroup analyses? This part is not a deal breaker to me - however - I believe that accounting for this in the protocol will increase the quality of the manuscript. "How will you account for different kind of meniscal tears? The way I read it, all specific meniscal treatments will be pooled despite the population. A meniscal tear in a young and active sports player will definitely behave differently than a degenerative one in a 70y old. What to do with the interpretability? And how will you deal with heterogeneity?"
---

REVIEWER	Kenneth Pihl Lund University Sweden
REVIEW RETURNED	18-Apr-2020

GENERAL COMMENTS	Thank you for the opportunity to re-review this interesting protocol manuscript by Ahmed et al. entitled “The Meniscal tear outcome (METRO) review: A protocol for a systematic review summarising the clinical course and patient experiences of meniscal tears in the current literature”. The authors have done a great job revising and addressing the concerns by me and the other reviewer. Still, I am not fully convinced that the quantitative part of the review will provide much new knowledge, as the clinical course, to my opinion, of different treatments have been widely reported previously. Only the part of the natural history of meniscal tears indeed needs more attention, however it is relatively unlikely that the review will provide much new here as the studies investigating this is extremely scarce, which a study by Abram et al. with one of the authors of the current study also showed. But of course, the lack of studies investigating that will be a result of itself. The qualitative part, however, is much more novel and an important field to explore. Maybe that should be the main focus of the review. Besides the above-mentioned thoughts regarding the purpose of the review, I only have few minor comments – see below. Introduction: Page 4, line 5: It is not fully clear if the incidence is annual or something else? Method: Page 5, line 56: Only prospective studies that include patients being treated for a meniscal tear – why exclude if any observational studies where no treatment is provided if the natural course is wanted? Page 7, line 5: Please elaborate on how the studies’ risk of bias is assessed – unclear if the “adequate, partially adequate, or inadequate” is a common method? Also, the reference (no 25) is for risk of bias assessment of reviews, not single studies – please provide a valid reference for the method. Page 7, line 40: Will all non-operative interventions be combined or reported separately as for the surgical treatments? That assumes that e.g. injections and exercise have comparable effect. Also, it might be worth extracting additional interventions such as if they receive any rehabilitation (e.g. after surgery, sham-surgery). Results: Page 8, line 41: As noted for the methods, will all non-operative interventions be combined or reported separately as for the surgical treatments? Reason to believe that effects are comparable between all non-surgical treatments?
---

	Page 8, line 50: This sentence is a bit confusing to me, as you have just stated that you will compare interventions if sufficient new studies? Discussion: Page 9, line 38_ To my knowledge, none of the two reviews cited included studies on meniscal repair. Overall, I acknowledge the great job the authors have done revising the manuscript, which concern an important topic. The major threats to the quantitative part however, is the risk of lacking studies that include patients not receiving any treatment, thus they will be unable to spread light on the natural course of meniscal tears.
--	--

VERSION 2 – AUTHOR RESPONSE

Reviewer: 1

Reviewer Name

Victor van de Graaf

Institution and Country

OLVG Amsterdam, the Netherlands

Please state any competing interests or state 'None declared':

None declared

Please leave your comments for the authors below

Clear and satisfactory revision of the manuscript. I only have minor issue, that is for the interpretation of the results. As stated in my previous revision, how will the authors deal with the type of tear? Are you planning on subgroup analyses?

This part is not a deal breaker to me - however - I believe that accounting for this in the protocol will increase the quality of the manuscript.

"How will you account for different kind of meniscal tears? The way I read it, all specific meniscal treatments will be pooled despite the population. A meniscal tear in a young and active sports player will definitely behave differently than a degenerative one in a 70y old. What to do with the interpretability? And how will you deal with heterogeneity?"

Thank you for your comment and review of our manuscript.

We have added a paragraph in the statistical analysis section. Where the studies report data for patients with OA and without OA, or presence or absence of traumatic mechanism, we aim to perform a subgroup analysis. This will allow exploration of variation in outcome over time in these different patient populations.

This will help account for the heterogeneity in study populations within the studies, which we will endeavour to describe narratively as best as we are able. Statistical heterogeneity will be assessed using the I² statistic.

Reviewer: 2

Reviewer Name

Kenneth Pihl

Institution and Country

Lund University
Sweden

Please state any competing interests or state 'None declared':
No conflicts of interests.

Please leave your comments for the authors below

Thank you for the opportunity to re-review this interesting protocol manuscript by Ahmed et al. entitled "The Meniscal tear outcome (METRO) review: A protocol for a systematic review summarising the clinical course and patient experiences of meniscal tears in the current literature".

The authors have done a great job revising and addressing the concerns by me and the other reviewer. Still, I am not fully convinced that the quantitative part of the review will provide much new knowledge, as the clinical course, to my opinion, of different treatments have been widely reported previously. Only the part of the natural history of meniscal tears indeed needs more attention, however it is relatively unlikely that the review will provide much new here as the studies investigating this is extremely scarce, which a study by Abram et al. with one of the authors of the current study also showed. But of course, the lack of studies investigating that will be a result of itself.

The qualitative part, however, is much more novel and an important field to explore. Maybe that should be the main focus of the review.

Thank you for you comments and review of the manuscript. We have aimed to produce this manuscript placing equal importance on each section. A scoping review did not identify many qualitative studies, which is an important and interesting finding to report nonetheless. As a result, we decided to keep the focus equal between the two. If however, the qualitative review reports interesting findings we will take on your advice and ensure it is reported as an important component of the final review.

With regards to the quantitative review, we agree that there has been a recent review, however, we believe our review is different and will add to the evidence base. Firstly, we aim to include non-randomised studies which were not included in the review by Abram et al. Secondly, our methodological design is different to that review. Instead of focussing on clinical effectiveness of each treatment we are focussing on variation in PROMS over time to see the patterns which emerge. This will allow healthcare providers to understand the clinical course of a condition to

plan treatment decisions. It will also allow researchers to plan future trials and timing of outcome assessment. In addition to this, we aim to include evidence from studies reporting outcomes from meniscal repairs as well as meniscectomies.

A scoping review for potential new studies for inclusion has identified the following studies in addition to the ones included in the Abram review:

Thorlund Jonas

Bloch, Englund Martin, Christensen Robin, Nissen Nis, Pihl Kenneth, Jørgensen Uffe et al. Patient reported outcomes in patients undergoing arthroscopic partial meniscectomy for traumatic or degenerative meniscal tears: comparative prospective cohort study *BMJ* 2017; 356 :j356

Berg B, Roos EM, Englund M, Kise NJ et al. Development of osteoarthritis in patients with degenerative meniscal tears treated with exercise therapy or surgery: a randomized controlled trial. Osteoarthritis Cartilage. 2020 Mar 14. pii: S1063-4584(20)30921-3. doi: 10.1016/j.joca.2020.01.020

Thaunat M, Fournier G, O'Loughlin P, Kouevidjin BT, Clowez G, Borella M, et al. Clinical outcome and failure analysis of medial meniscus bucket-handle tear repair: a series of 96 patients with a minimum 2 year follow-up. *Archives of Orthopaedic & Trauma Surgery*. 2020;28:28.

Saltzman BM, Cotter EJ, Wang KC, Rice R, Manning BT, Yanke AB, et al. Arthroscopically Repaired Bucket-Handle Meniscus Tears: Patient Demographics, Postoperative Outcomes, and a Comparison of Success and Failure Cases. *Cartilage*. 2020;11(1):77-87.

Pihl K, Ensor J, Peat G, Englund M, Lohmander S, Jørgensen U, et al. Wild goose chase - no predictable patient subgroups benefit from meniscal surgery: patient-reported outcomes of 641 patients 1 year after surgery. *British Journal of Sports Medicine*. 2020;54(1):13-22.

These are a few of the additional studies we have found. We believe there is enough new evidence there to warrant an updated search. We are also employing a different methodology and inclusion criteria (Non- RCTs) which will ensure this study will add to the evidence base.

Besides the above-mentioned thoughts regarding the purpose of the review, I only have few minor comments – see below.

Introduction:

Page 4, line 5: It is not fully clear if the incidence is annual or something else?

This incidence is annual. 60-70 per 100,000 population-years. We have corrected this.

Method:

Page 5, line 56: Only prospective studies that include patients being treated for a meniscal tear – why exclude if any observational studies where no treatment is provided if the natural course is wanted?

The principal reason retrospective studies were excluded is that they are subject to recall bias. Prospective studies with a well-defined eligibility criteria have been used in previous studies on natural history. This why we excluded retrospective studies.

Page 7, line 5: Please elaborate on how the studies' risk of bias is assessed – unclear if the “adequate, partially adequate, or inadequate” is a common method? Also, the reference (no 25) is for risk of bias assessment of reviews, not single studies – please provide a valid reference for the method.

Quality assessment will be carried out using the CASP tool. Based on the assessment two review authors will determine whether the quality of the study is “adequate, partially adequate, or inadequate”. If there are any discrepancies a senior author will be consulted. If a study is deemed inadequate it will not be included in the meta-synthesis. A valid reference has been provided.

Page 7, line 40: Will all non-operative interventions be combined or reported separately as for the surgical treatments? That assumes that e.g. injections and exercise have comparable effect. Also, it might be worth extracting additional interventions such as if they receive any rehabilitation (e.g. after surgery, sham-surgery).

Our plan is to combine all non-operative interventions separately. If there is sufficient studies we will perform a subgroup analysis where we will pool the results of exercise therapy and pharmacological interventions separately.

Results:

Page 8, line 41: As noted for the methods, will all non-operative interventions be combined or reported separately as for the surgical treatments? Reason to believe that effects are comparable between all non-surgical treatments?

Our plan is to combine all non-operative interventions separately. If there is sufficient studies we will perform a subgroup analysis where we will pool the results of exercise therapy and pharmacological interventions separately.

Page 8, line 50: This sentence is a bit confusing to me, as you have just stated that you will compare interventions if sufficient new studies?

This sentence has been reviewed. The main objective of this review is to summarise the short, medium and long-term temporal patterns of change in patient reported outcome measures (PROMs) in patients with a meniscal tear regardless of treatment. However, as we are planning on including non-randomised studies and new studies we will also assess the clinical effectiveness of operative interventions in comparison to non-operative interventions using the methods described in this paragraph.

Discussion:

Page 9, line 38_ To my knowledge, none of the two reviews cited included studies on meniscal repair.

Thank you for correcting this. This was placed in error and the sentence has been corrected.

Overall, I acknowledge the great job the authors have done revising the manuscript, which concern an important topic. The major threats to the quantitative part however, is the risk of lacking studies that include patients not receiving any treatment, thus they will be unable to spread light on the natural course of meniscal tears.

Thank you for the helpful comments. The question about insufficient data on natural history could be the case, but needs a good quality updated formal review to either document that fact or refute it. Even then, documenting the current evidence on progression over time will benefit both researchers planning further trials and patients who wish to understand their expectations of outcome. We therefore believe that this comprehensive review will be of value.

VERSION 3 – REVIEW

REVIEWER	Victor van de Graaf OLVG Amsterdam, The Netherlands
REVIEW RETURNED	31-May-2020

GENERAL COMMENTS	No further comments. Well done job by the team.
---

REVIEWER	Kenneth Pihl Lund University Clinical Epidemiology Unit Sweden
REVIEW RETURNED	22-Jun-2020

GENERAL COMMENTS	The authors have done a great job revising the manuscript. I have no further comments.
--